# Methodological review to develop a list of bias items for adaptive clinical trials: Protocol and rationale

Phillip Staibano [1,2]*, Tyler McKechnie[2,3], Alex Thabane[2], Daniel Olteanu[4], Keean Nanji[2,5], Han Zhang[1], Carole Lunny[6], Michael Au[1], Michael K. Gupta[1], Jesse D. Pasternak[7], Sameer Parpia[2], JEM (Ted) Young[1], Mohit Bhandari[2,8]

1 Division of Otolaryngology–Head and Neck Surgery, Department of Surgery, McMaster University, Hamilton, Ontario, Canada, 2 Department of Health Research Methodology, Evidence, and Impact, McMaster University, Hamilton, Ontario, Canada, 3 Division of General Surgery, Department of Surgery, McMaster University, Hamilton, Ontario, Canada, 4 Michael G. DeGroote School of Medicine, Hamilton, Ontario, Canada, 5 Division of Ophthalmology, Department of Surgery, McMaster University, Hamilton, Ontario, Canada, 6 Therapeutics Initiative, SPOR Evidence Alliance, Society for Research Synthesis Methods, Cochrane Bias Methods Group, Cochrane Hypertension Review Group at the University of British Columbia, Vancouver, Canada, 7 Endocrine Surgery Section Head, Division of General Surgery, Department of Surgery, University Health Network, University of Toronto, Toronto, Ontario, Canada, 8 Division of Orthopedic Surgery, Department of Surgery, McMaster University, Hamilton, Ontario, Canada

* phillip.staibano@medportal.ca

**Data Availability Statement:** No datasets were generated or analysed during the current study. All relevant data from this study will be made available upon study completion.

## Abstract

### Background

Randomized-clinical trials (RCTs) are the gold-standard for comparing health care interventions, but can be limited by early termination, feasibility issues, and prolonged time to trial reporting. Adaptive clinical trials (ACTs), which are defined by pre-planned modifications and analyses that occur after starting patient recruitment, are gaining popularity as they can streamline trial design and time to reporting. As adaptive methodologies continue to be adopted by researchers, it will be critical to develop a risk of bias tool that evaluates the unique methodological features of ACTs so that their quality can be improved and standardized for the future. In our proposed methodological review, we will develop a list of risk of bias items and concepts, so that a risk of bias tool specific to ACTs can be developed.

### Methods and analysis

We will perform a systematic database search to capture studies that have proposed or reviewed items pertaining to methodological risk, bias, and/or quality in ACTs. We will perform a comprehensive search of citation databases, such as Ovid MEDLINE, EMBASE, CENTRAL, the Cochrane library, and Web of Science, in addition to multiple grey literature sources to capture published and unpublished literature related to studies evaluating the methodological quality of ACTs. We will also search methodological registries for any risk of bias tools for ACTs. All screening and review stages will be performed in duplicate with a third senior author serving as arbitrator for any discrepancies. For all studies of methodological quality and risk of bias, we will extract all pertinent bias items, concepts, and/or tools. We

**Funding:** The author(s) received no specific funding for this work.

**Competing interests:** The authors have declared that no competing interests exist.

will combine conceptually similar items in a descriptive manner and classify them as referring to bias or to other aspects of methodological quality, such as reporting. We will plan to generate pertinent risk of bias items to generate a candidate tool that will undergo further refinement, testing, and validation in future development stages.

## Ethics and dissemination

This review does not require ethics approval as human subjects are not involved. As mentioned previously, this study is the first step in developing a tool to evaluate the risk of bias and methodological quality of ACTs. The findings of this review will inform a Delphi study and the development of a risk of bias tool for ACTs. We plan on publishing this review in a peer-reviewed journal and to present these findings at international scientific conferences.

## Introduction

Evidence-based medicine has revolutionized the development of clinical practice guidelines and decision making in healthcare [1]. Randomized-controlled trials (RCTs) are the gold-standard for comparing the effectiveness and safety of novel healthcare interventions [2]. Conventional RCTs, however, can be burdened by high costs, early termination due to feasibility issues, and an overly rigid design that does not permit adjustments for unforeseen challenges [3]. These issues are amplified in surgical trials and as such, the annual number of published surgical trials remains stagnant [4, 5]. As a response to these challenges, researchers have begun using adaptive trial designs, which allow for dynamic protocol changes after beginning patient recruitment. Adaptive clinical trials (ACTs) utilize at least one pre-planned interim analysis to modify the protocol of an ongoing trial while maintaining integrity and validity of the data collected [6]. Trial adaptations performed following an interim analysis include sample size re-calculation, editing the number of treatment arms, amending allocation ratios, and/ or terminating a trial early for success or lack of efficacy. Adaptive designs can improve the trial running process by optimizing patient recruitment, combining clinical trial stages, minimizing sample size, and accelerating time to trial analysis and reporting [6]. For instance, the TAILoR trial of telmisartan in HIV employed an interim analysis at half maximal patient recruitment and dropped the most ineffective medication dosage group based on a pre-specified efficacy threshold [7]. Moreover, adaptive designs streamlined the clinical trial process during COVID-19 by optimizing the number of therapies evaluated and minimizing the number of patients enrolled in each trial [8]. Adopting ACTs when prolonged RCTs are impractical may also reduce necessary funding, thereby overcoming barriers to conducting trials in developing nations [9]. Stakeholders, however, report that adaptive trial designs remain nebulous with practical barriers, including high bias potential, ethical concerns, and a lack of knowledge dissemination amongst trialists [10]. Other challenges include the need for a robust network of researchers and biostatisticians to ensure that the, often complex, trial protocol is well-planned and has undergone rigorous statistical stimulation prior to beginning patient recruitment [11]. In addition, statistical software required for adaptive methods is limited in its accessibility and expensive. Clinicians, researchers, and funding agencies are not well-versed in adaptive design terminology and practices, nor is there proper standardization in adaptive trial reporting [12].

In 2020, CONSORT published an extension for adaptive trials to guide ACT reporting [13]. These guidelines ACT-specific methodological components such as pre-planned interim

**Table 1. Tools and checklists to aid in randomized controlled trial methodological quality.** Adapted from Lunny et al. (2021) [31].

| Tool purpose | Examples of tools or checklists | Description of the example tool or checklist | Available tool for ACTs |
|---|---|---|---|
| **Assess the quality of published RCTs** | CASP-RCT [24], CEBM-RCT [16], Jadad scale [15], JBI checklist [25], LEGEND [27], TRACT [17], SIGN [26] | These tools to assist in conducting and reviewing published RCTs that consist of multiple domains addressing trial design and internal validity, result reporting, and interpretability. | None |
| **Assess the risk of bias of published RCTs** | Cochrane RoB tool (2.0) [14] | Cochrane RoB 2.0 is a five-domain domain tool that evaluates bias arising from randomization, protocol deviations, missing outcome data, outcome measurement, and result reporting. Studies are given an overall bias risk score of low, high, or some concerns. | None |
| **Assess the external validity of RCTs** | None [20] | None | None |
| **Guidance for the complete reporting of RCTs** | CONSORT [21] | CONSORT is a 25-item checklist that assists with transparent reporting of RCT design, analysis, and interpretation. | CONSORT-ACE [13] |
| **Guidance checklist for clinical readers of RCTs** | Godin et al. (2011) [23], Nimavat et al. (2020) [22] | These resources provide a short checklist of simple questions addressing methodology, reporting, and interpretation for clinicians to keep in mind when appraising RCTs. | Ferreira et al. (2020) [18], Park et al. (2020) [19] |

CASP-RCT, critical appraisal skills program–randomized controlled trials; CEBM-RCT, Centre for Evidence-Based Medicine–randomized controlled trials; CONSORT, consolidated standards of reporting trials; CONSORT-ACE, consolidated standards of reporting trials adaptive designs extension; JBI, Joanne Briggs Institute; LEGEND, Let Evidence Guide Every New Decision; RCT, randomized controlled trial; RoB, risk of bias; SIGN, Scottish Intercollegiate Guidelines, Network; TRACT, Trustworthiness in randomised controlled trials

analyses and sample size estimation (and re-estimation) descriptions [13]. In conjunction with CONSORT reporting guidelines, a validated risk of bias tool developed in a similar manner to the Cochrane risk-of-bias 2.0 tool [14], may improve the design of ACTs and the quality of future meta-analyses combing ACTs. Risk of bias tools are designed for specific study designs (e.g. RCTs) and help to promote methodological transparency and reproducibility while minimizing bias, so results can be accurately interpreted and soundly applied to patient care. For conventional RCTs, there exist several tools and checklists to guide reporting or evaluate quality and risk of bias (Table 1) [14–27]. There is no existing risk of bias tool to evaluate the methodological limitations of ACTs, which is of particular importance due to the potential for ACTs to be impacted by bias if not soundly designed [28]. It is for this reason that we have decided to embark upon creating a novel risk of bias tool to improve the quality of future ACTs and meta-analyses of ACTs.

Our proposed methodological review has two main objectives: (1) to identify and describe any current risk of bias items, tools, or checklists specific for ACTs and (2) to compile a list of risk of bias items and concepts that can be used to develop a risk of bias tool for ACTs. We will develop our risk of bias tool for ACTs in accordance with the framework described by Whiting et al. (2017) [29].

## Materials and methods

### Study design

We present a protocol to describe the rationale for performing a methodological review of ACTs and generate a list of risk of bias items and concepts related to ACTs. We will follow the methodological framework proposed by Whiting et al. (2017) and Sanderson et al. (2007) [29, 30]. This protocol was written with guidance from a methodology review protocol published by Lunny and colleagues, who set out to create a novel risk of bias tool for network meta-analyses [31]. As described by Lunny and colleagues, subsequent steps in creating a risk of bias tool

will include (1) a Delphi survey and panel to select, refine, and compile bias items into a single candidate tool; (2) a pilot test to further refine the proposed tool; and (3) a knowledge translation strategy to disseminate the final risk of bias tool [31]. With regards to the Delphi study, we will plan to first distribute a knowledge survey to methodologists and content experts to gather their opinion on an ACT risk of bias tool and how it should be utilized and disseminated. Next, a pre-selected steering panel will generate a candidate list of risk of bias items that will be distributed in multiple rounds to a Delphi panel that will rate the utility of including each item and/or concept in the candidate tool. Pilot testing will include the evaluation of tool useability, efficiency, and comprehension among content experts in adaptive methodologies and trial design. Our final knowledge translation strategy will include publication and presentation of the final tool, housing the tool in an accessible website, and providing training sessions and webinars to future tool users [29]. These steps will be further addressed in future studies as we progress through this framework in developing this proposed ACT risk of bias tool. We did perform this methodological review protocol in accordance with the PRISMA-P checklist [32].

## Eligibility

There will be two types of studies included in this methodological review (Table 2). *Study type 1* will be studies that describe items and/or concepts related to bias, reporting, or methodological quality of ACTs. We will retain all items related to methodological bias and/or reporting as they may be able to be translated into a risk-of-bias tool. *Study type 2* will be studies that assess the methodological quality, or risk of bias, of ACTs using criteria that focus on methodological features specific to ACTs. Both study types will be analyzed with the goal of collating bias items and/or concepts. We will also review and gather related items/concepts from any published risk of bias tools or reporting quality tools used for conventional RCTs (Table 1). We

**Table 2. Eligible study types for methodological review.**

**Study type 1:** Studies that describe items and concepts relating to bias, reporting, and/or methodological quality in adaptive clinical trials

| Inclusion criteria | Exclusion criteria |
|---|---|
| • Any publication type (e.g., primary study, published abstract, thesis dissertation, pre-print manuscript)<br>• Published in any language.<br>• Any study design (i.e., randomized trial, meta-analysis, scoping review, systematic review, narrative review, author editorial, letter to the editor).<br>• Mention, describe, evaluate at least one risk of bias, reporting, or methodological quality item, concept, tool, or checklist pertaining to ACTs. | • Reported methodological item or tool not mentioned or described in the context of adaptive trial designs. |

**Study type 2:** Studies that assess the methodological quality or risk of bias of adaptive clinical trials

| Inclusion criteria | Exclusion criteria |
|---|---|
| • Any publication type (e.g., primary study, published abstract, thesis dissertation, pre-print manuscript)<br>• Published in any language.<br>• Any study design (i.e., randomized trial, meta-analysis, scoping review, systematic review, narrative review, author editorial, letter to the editor).<br>• Evaluate or describe the methodological quality or risk of bias of at least one ACT | • Reported methodological item or tool not mentioned or described in the context of adaptive trial designs. |

ACT, adaptive clinical trials

will include all articles with any publication status and written in any language. We will focus on methodological studies of ACTs and so, we will not be evaluating published ACTs or studies based on disease type, clinical populations, or tested interventions. In cases where the co-authors are not fluent or review authors are unable to understand the study text, we will utilize Google Translate (Mountain View, CA, USA).

## Search strategy

All databases to be used in this review were selected with guidance from Lunny et al. (2021) [31]. We will search all databases with no language or publication type limits. We will search the following databases: MEDLINE (Ovid), CINAHL, EMBASE (Ovid), the Cochrane library, the Cochrane Central Register of Controlled Trials (CENTRAL), Web of Science, BIOSIS, Derwent Innovations Database, and KCI. We will also search clinical trials registries including clinicaltrials.gov and the WHO International Clinical Trials Registry Platform (ICTRP). We will search the following grey literature databases and resources: the EQUATOR network, dissertation abstracts, websites of evidence synthesis organizations (e.g., Campbell Collaboration, Cochrane Multiple Treatments Group, CADTH, NICE-DSU, Health Technology Assessment International (HTAi), Pharmaceutical Benefits Advisory Committee, Institut für Qualität und Wirtschaftlichkeit im Gesundheitswesen, European Network for Health Technology Assessment, Guidelines International Network, ISPOR, International Network of Agencies for Health Technology Assessment, and JBI), and methods collections (e.g., Cochrane Methodology Register, AHRQ Effective Healthcare Programme). We will also search LIGHTS and LATITUDES (https://www.latitudes-network.org/), which are two methodological registries that capture guidance and validity assessment tools, respectively [33]. All online registries will be searched using the following terms "adaptive clinical trial", "bias", and/or risk of bias". The words found within the titles, abstracts, and MeSH terms of relevant articles were used to develop focused search strategies for each database. Reference lists of studies found will also be searched for additional papers to be included. The MEDLINE search will be validated for 10 studies identified by senior authors prior to screening. Eligibility screening will only begin after these 10 trials are identified from the search strategy. All database search strategies are described in S1 File.

The search strategy will be generated by two authors (P.S. and D.O.) alongside a librarian specialist. It will be generated and reviewed in accordance with PRESS (Peer Review Electronic Search Strategies) guidelines [34]. Any concerns with search strategy generation will be raised with a senior methodologist (M.B.). The database search will be conducted without limitations to publication type, status, language, or date to identify existing tools or articles.

## Screening and data extraction

First, we will pilot eligibility criteria in Microsoft Excel (Redmond, WA, USA) by evaluating a sample of 25 citations amongst two independent reviewers. If high agreement is achieved (≥70%), then we will continue to abstract screening with two reviewers. If less than high agreement is achieved, then the eligibility criteria will be re-examined, and additional teaching sessions will be provided to reviewers. All screening and full-text review will be conducted using the web-based application, Covidence (http://www.covidence.org; Melbourne, Australia). Study titles and abstracts will then be assessed for relevance and eligibility. All screening and full-text review will be performed in accordance with PRISMA guidelines (S2 File). Any disagreements identified during these screening and review stages will be resolved via discussion until consensus is reached. A third senior reviewer will arbitrate if screening or full-text review disagreements cannot be resolved.

A data extraction form will be generated using Microsoft Excel and piloted by reviewers on five included studies. Two authors will extract data from all included studies. We will first categorize all sources based on our eligibility criteria and we will extract author details, publication year, and study type. For all studies that identified bias items, tools, or quantified methodological bias in ACTs, we will generate the following list of headings: type of tool (e.g., tool, scale, checklist, or domain-based tool); scope of the tool; number of items within the tool; domains within the tool; whether the item relates to reporting or methodological quality; ratings of items and domains within the tool; methods used to develop the tool and the availability of an "explanation and elaboration". These fields were all derived from Lunny et al. (2021) and Page et al. (2018) [31, 35]. Data will be extracted on items that are relevant to ACTs and all items will initially be extracted verbatim.

## Data analysis and reporting

All studies evaluating methodological quality, and/or proposing bias tools, items, concepts, checklists will be collated. These studies will undergo descriptive analyses based upon previously extracted fields. All bias items will all be mapped to corresponding domains within the CONSORT-ACE guidelines, as this is the only known quality tool specific to ACT features. If no risk-of-bias tools or relevant items are identified in our review, then we will plan to hypothesize items and formulate a candidate tool. We will develop a candidate tool that will undergo refinement by the authors, and in subsequent steps will be further evaluated and refined using a Delphi consensus method prior to undergoing validation and testing. Currently, our plan is to create a standalone risk of bias tool for ACTs that will gather inspiration from the domains of the Cochrane RoB 2.0 tool. But, as we proceed to the knowledge user survey, Delphi process, and candidate tool development phases, we will further evaluate the feasibility and advantages of instead developing an extension to the Cochrane RoB 2.0. All statistical analyses will be performed using R software (version 4.3.2, Vienna, Austria).

## Public and public involvement

Patient or the public were not involved in the design of this research protocol.

## Discussion

A comprehensive risk of bias tool is needed to improve the reproducibility and transparency of future ACTs and ACT meta-analyses. A recent scoping review demonstrated that ACTs adhere poorly to the reporting recommendations published in the CONSORT-ACE statement [12]. We are, therefore, going to use the framework developed by Whiting and colleagues to develop a new risk of bias tool to improve the quality of ACTs [29]. A risk of bias tool for ACTs is needed since these designs often demonstrate key methodological differences from conventional RCTs, such as interim analyses and adaptive decisions made after the trial has begun patient recruitment [6]. The current repertoire of risk of bias tools used for conventional RCTs do not possess domains that address these unique adaptive design features. Moreover, adaptive decisions made during the running of a clinical trial can increase methodological bias in the final trial analysis, thus emphasizing the need for a risk of bias tool to be used when evaluating ACTs. Potential limitations include missing any published methodology studies or ROB tools of ACTs, but we will counteract this via a broad search strategy of peer-reviewed literature databases, grey literature databases, and methodological tool repositories. As computational technologies continue to improve, their role in generating adaptive trial paradigms and performing statistical simulations may revolutionize the future of trial design and medical innovation [36]. We, must, therefore ensure that methodological tools are

developed at a similar pace, so that these novel trial designs are standardized, transparent, reproducible, and interpretable for the future.

## Supporting information

**S1 File. Search strategies.**
(DOCX)

**S2 File. PRISMA flow diagram for prospective article screening and full-text review.**
(DOCX)

**S3 File. Completed PRISMA-P checklist.**
(DOC)

## Author Contributions

**Conceptualization:** Phillip Staibano, Tyler McKechnie, Alex Thabane, Keean Nanji, Han Zhang, Carole Lunny, Michael Au, Michael K. Gupta, Jesse D. Pasternak, Sameer Parpia, JEM (Ted) Young, Mohit Bhandari.

**Data curation:** Tyler McKechnie, Han Zhang.

**Methodology:** Phillip Staibano, Tyler McKechnie, Alex Thabane, Daniel Olteanu, Keean Nanji, Han Zhang, Carole Lunny, Michael Au, Michael K. Gupta, Jesse D. Pasternak.

**Project administration:** Phillip Staibano.

**Resources:** Phillip Staibano.

**Software:** Phillip Staibano.

**Validation:** Alex Thabane.

**Writing – original draft:** Phillip Staibano, Tyler McKechnie, Alex Thabane, Daniel Olteanu, Keean Nanji, Han Zhang, Carole Lunny, Michael Au, Jesse D. Pasternak, Sameer Parpia, JEM (Ted) Young, Mohit Bhandari.

**Writing – review & editing:** Phillip Staibano, Tyler McKechnie, Alex Thabane, Daniel Olteanu, Keean Nanji, Han Zhang, Carole Lunny, Michael Au, Michael K. Gupta, Jesse D. Pasternak, Sameer Parpia, JEM (Ted) Young, Mohit Bhandari.

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
