## [Decision Letter · Decision Letter 0]

4 Jul 2024

PONE-D-24-13528Methodological review to develop a list of bias items for adaptive clinical trials: Protocol and rationalePLOS ONE

Dear Dr. Staibano,

Thank you for submitting your manuscript to PLOS ONE. After careful consideration, we feel that it has merit but does not fully meet PLOS ONE’s publication criteria as it currently stands. Therefore, we invite you to submit a revised version of the manuscript that addresses the points raised during the review process.

We look forward to receiving your revised manuscript.

Kind regards,

Sathish Muthu

Academic Editor

PLOS ONE

Additional Editor Comments (if provided):

I appreciate the effort put forth in this work and the importance of developing a risk-of-bias tool for adaptive clinical trials (ACTs). This is a crucial step towards improving the quality and transparency of ACTs, which have the potential to streamline clinical research. I would the authors to provide a clear view of the Adaptive clinical trials for the benefit of the authors in the introduction with examples and their advantages and limitations. Further in the methodological aspect does the authors consider the tool as a standalone tool or as a additive tool to RoB2 is not clearly made out. Analysis need to be detailed on the type of regression to be utilized.

Reviewers' comments:

Reviewer's Responses to Questions

**Comments to the Author**

1. Does the manuscript provide a valid rationale for the proposed study, with clearly identified and justified research questions?

Reviewer #1: Yes

2. Is the protocol technically sound and planned in a manner that will lead to a meaningful outcome and allow testing the stated hypotheses?

Reviewer #1: Yes

3. Is the methodology feasible and described in sufficient detail to allow the work to be replicable?

Reviewer #1: Yes

4. Have the authors described where all data underlying the findings will be made available when the study is complete?

Reviewer #1: No

5. Is the manuscript presented in an intelligible fashion and written in standard English?

Reviewer #1: Yes

6. Review Comments to the Author

You may also provide optional suggestions and comments to authors that they might find helpful in planning their study.

Reviewer #1: Thank you for the opportunity to review your manuscript entitled "Methodological review to develop a list of bias items for adaptive clinical trials: Protocol and rationale." I appreciate the effort put forth in this work and the importance of developing a risk-of-bias tool for adaptive clinical trials (ACTs). This is a crucial step towards improving the quality and transparency of ACTs, which have the potential to streamline clinical research. I hope the following comments and suggestions will improve the quality of the manuscript.

Abstract

1. The authors mentioned that "we will perform regression analysis to identify factors associated with poor reporting quality and high risk of bias," but this was not discussed in the methods section in the main text. Please specify the type of regression analysis to be used (e.g., Poisson, linear, logistic) and provide details on the variables to be included in the regression model in the main text.

2. It would be helpful to provide more details on the planned dissemination strategy (e.g., publication, conference presentation, or any other strategies) for the developed risk-of-bias tool, beyond mentioning that this is the first step in its development.

Introduction

3. The description of Adaptive clinical trials (ACTs) does not provide a clear picture of the concept. Please consider explaining the concept in this section to facilitate better understanding of the later sections.

4. Adaptive clinical trials (ACTs) is mentioned twice in one sentence: "Adaptive clinical trials (ACTs) Adaptive clinical trial (ACTs)." Please revise for better clarity.

5. The authors may consider briefly mentioning the potential challenges or limitations of ACTs, in addition to the advantages, to further highlight the importance of a risk-of-bias tool.

Methods

6. Please consider providing a brief overview or example of the Delphi process and subsequent steps planned for the development and validation of the risk-of-bias tool.

7. It is unclear how prospective/retrospective cohort, cross-sectional, and case series studies could provide information related to methodological bias, reporting, or quality in ACTs. Please clarify or revise the inclusion criteria under: "Study type 2: Studies that describe items relating to methodological bias, reporting, or quality in ACTs."

8. The authors mentioned "Eligibility: There will be three types of studies included in this scoping review." Did the authors mean methodological review instead of scoping review?

9. Please clarify whether the authors plan to include studies that assess the methodological quality or risk of bias of ACTs using tools designed for traditional RCTs (e.g., Cochrane RoB 2.0, Jadad scale), or if the focus is solely on tools specifically designed for ACTs.

Discussion

10. Please expand on the potential implications and impact of developing a risk-of-bias tool for ACTs, both for researchers and clinicians.

7. PLOS authors have the option to publish the peer review history of their article (what does this mean?). If published, this will include your full peer review and any attached files.

Reviewer #1: No

---

## [Author Response · Author response to Decision Letter 0]

4 Jul 2024

PONE-D-24-13528

Methodological review to develop a list of bias items for adaptive clinical trials: Protocol and rationale

PLOS ONE

Dear Dr. Staibano,

Thank you for submitting your manuscript to PLOS ONE. After careful consideration, we feel that it has merit but does not fully meet PLOS ONE’s publication criteria as it currently stands. Therefore, we invite you to submit a revised version of the manuscript that addresses the points raised during the review process.

We look forward to receiving your revised manuscript.

Kind regards,

Sathish Muthu

Academic Editor

PLOS ONE

Additional Editor Comments (if provided):

I appreciate the effort put forth in this work and the importance of developing a risk-of-bias tool for adaptive clinical trials (ACTs). This is a crucial step towards improving the quality and transparency of ACTs, which have the potential to streamline clinical research. I would the authors to provide a clear view of the Adaptive clinical trials for the benefit of the authors in the introduction with examples and their advantages and limitations. Further in the methodological aspect does the authors consider the tool as a standalone tool or as a additive tool to RoB2 is not clearly made out. Analysis need to be detailed on the type of regression to be utilized.

We have removed the regression analysis, as we do not feel as though it contributes to development of a risk of bias tool for ACTs. We stated in the methods section that we will plan to create a standalone tool, but as the process enters the Delphi and candidate tool development stages, we will determine the role of instead creating an extension to the Cochrane RoB 2.0 tool. 

Comments to the Author

1. Does the manuscript provide a valid rationale for the proposed study, with clearly identified and justified research questions?

Reviewer #1: Yes

2. Is the protocol technically sound and planned in a manner that will lead to a meaningful outcome and allow testing the stated hypotheses?

Reviewer #1: Yes

3. Is the methodology feasible and described in sufficient detail to allow the work to be replicable?

Reviewer #1: Yes

4. Have the authors described where all data underlying the findings will be made available when the study is complete?

Reviewer #1: No

5. Is the manuscript presented in an intelligible fashion and written in standard English?

Reviewer #1: Yes

6. Review Comments to the Author

You may also provide optional suggestions and comments to authors that they might find helpful in planning their study.

Reviewer #1: Thank you for the opportunity to review your manuscript entitled "Methodological review to develop a list of bias items for adaptive clinical trials: Protocol and rationale." I appreciate the effort put forth in this work and the importance of developing a risk-of-bias tool for adaptive clinical trials (ACTs). This is a crucial step towards improving the quality and transparency of ACTs, which have the potential to streamline clinical research. I hope the following comments and suggestions will improve the quality of the manuscript.

Abstract

1. The authors mentioned that "we will perform regression analysis to identify factors associated with poor reporting quality and high risk of bias," but this was not discussed in the methods section in the main text. Please specify the type of regression analysis to be used (e.g., Poisson, linear, logistic) and provide details on the variables to be included in the regression model in the main text.

Thank you. This was an oversight and was not removed from a previous version of the draft. We do not believe that a regression analysis will contribute to the development of a risk of bias tool. 

2. It would be helpful to provide more details on the planned dissemination strategy (e.g., publication, conference presentation, or any other strategies) for the developed risk-of-bias tool, beyond mentioning that this is the first step in its development.

We have added to this section. 

Introduction

3. The description of Adaptive clinical trials (ACTs) does not provide a clear picture of the concept. Please consider explaining the concept in this section to facilitate better understanding of the later sections.

Thank you. We further explained this concept. 

4. Adaptive clinical trials (ACTs) is mentioned twice in one sentence: "Adaptive clinical trials (ACTs) Adaptive clinical trial (ACTs)." Please revise for better clarity.

This has been amended. Thank you. 

5. The authors may consider briefly mentioning the potential challenges or limitations of ACTs, in addition to the advantages, to further highlight the importance of a risk-of-bias tool.

We have further added to the limitations of ACTs. Thank you. 

Methods

6. Please consider providing a brief overview or example of the Delphi process and subsequent steps planned for the development and validation of the risk-of-bias tool.

This has now been included. Thanks. 

7. It is unclear how prospective/retrospective cohort, cross-sectional, and case series studies could provide information related to methodological bias, reporting, or quality in ACTs. Please clarify or revise the inclusion criteria under: "Study type 2: Studies that describe items relating to methodological bias, reporting, or quality in ACTs."

We agree. Those study types will realistically not evaluate methodological quality in ACTs. We have thus changed this to only include study types that are likely to include this information. Further, we have amended this to only include two study types (1) studies that have described or suggested any risk of bias or methodological quality items for ACTs and (2) any existing studies that have evaluated methodological bias/quality in ACTs. We have removed those study types. We will also not be looking for any published ACTs since this will not contribute to the identification of risk of bias items. 

8. The authors mentioned "Eligibility: There will be three types of studies included in this scoping review." Did the authors mean methodological review instead of scoping review?

Yes, this was changed. Thank you. 

9. Please clarify whether the authors plan to include studies that assess the methodological quality or risk of bias of ACTs using tools designed for traditional RCTs (e.g., Cochrane RoB 2.0, Jadad scale), or if the focus is solely on tools specifically designed for ACTs.

We will include those tools as well. We have stated this more clearly. 

Discussion

10. Please expand on the potential implications and impact of developing a risk-of-bias tool for ACTs, both for researchers and clinicians.

We have done this. Thank you. 

7. PLOS authors have the option to publish the peer review history of their article (what does this mean?). If published, this will include your full peer review and any attached files.

Do you want your identity to be public for this peer review? For information about this choice, including consent withdrawal, please see our Privacy Policy.

Reviewer #1: No

---

## [Decision Letter · Decision Letter 1]

23 Aug 2024

PONE-D-24-13528R1Methodological review to develop a list of bias items for adaptive clinical trials: Protocol and rationalePLOS ONE

Dear Dr. Staibano,

Thank you for submitting your manuscript to PLOS ONE. After careful consideration, we feel that it has merit but does not fully meet PLOS ONE’s publication criteria as it currently stands. Therefore, we invite you to submit a revised version of the manuscript that addresses the points raised during the review process.

We look forward to receiving your revised manuscript.

Kind regards,

Sathish Muthu

Academic Editor

PLOS ONE

Journal Requirements:

Additional Editor Comments:

Kindly address the concerns and comments of the reviewers as noted below

The manuscript demonstrates a high level of quality and possesses considerable research significance; however, there remain several issues that require resolution, specifically as outlined below:

1. Abbreviations in the tables should be completed.

2. Since the screening literature does not limit the study content, the included trials will vary greatly in terms of study purpose, clinical stage, disease type, etc., and there may be large heterogeneity in the study results. How to solve this problem? Also, please add limitations of this study to the discussion.

3. Please supplement the flow diagram for the selection process of articles.

Reviewers' comments:

Reviewer's Responses to Questions

**Comments to the Author**

1. Does the manuscript provide a valid rationale for the proposed study, with clearly identified and justified research questions?

Reviewer #1: Yes

Reviewer #2: Yes

Reviewer #3: Partly

2. Is the protocol technically sound and planned in a manner that will lead to a meaningful outcome and allow testing the stated hypotheses?

Reviewer #1: Yes

Reviewer #2: Yes

Reviewer #3: Partly

3. Is the methodology feasible and described in sufficient detail to allow the work to be replicable?

Reviewer #1: Yes

Reviewer #2: Yes

Reviewer #3: Yes

4. Have the authors described where all data underlying the findings will be made available when the study is complete?

Reviewer #1: No

Reviewer #2: Yes

Reviewer #3: Yes

5. Is the manuscript presented in an intelligible fashion and written in standard English?

Reviewer #1: Yes

Reviewer #2: Yes

Reviewer #3: Yes

6. Review Comments to the Author

You may also provide optional suggestions and comments to authors that they might find helpful in planning their study.

Reviewer #1: The authors have thoroughly addressed my feedback in this version, and I appreciate the effort put into the revisions. It looks well-polished and complete. Thank you for your work on this.

Reviewer #2: I have been able to read the revised version of the article. The authors have made a significant effort to improve the manuscript. I congratulate the authors for this novel proposal.

Reviewer #3: The manuscript demonstrates a high level of quality and possesses considerable research significance; however, there remain several issues that require resolution, specifically as outlined below:

1. Abbreviations in the tables should be completed.

2. Since the screening literature does not limit the study content, the included trials will vary greatly in terms of study purpose, clinical stage, disease type, etc., and there may be large heterogeneity in the study results. How to solve this problem? Also, please add limitations of this study to the discussion.

3. Please supplement the flow diagram for the selection process of articles.

7. PLOS authors have the option to publish the peer review history of their article (what does this mean?). If published, this will include your full peer review and any attached files.

Reviewer #1: No

Reviewer #2: **Yes: **M. Pardo Rios

Reviewer #3: No

---

## [Author Response · Author response to Decision Letter 1]

23 Aug 2024

PONE-D-24-13528R1

Methodological review to develop a list of bias items for adaptive clinical trials: Protocol and rationale

PLOS ONE

Dear Dr. Staibano,

Thank you for submitting your manuscript to PLOS ONE. After careful consideration, we feel that it has merit but does not fully meet PLOS ONE’s publication criteria as it currently stands. Therefore, we invite you to submit a revised version of the manuscript that addresses the points raised during the review process.

We look forward to receiving your revised manuscript.

Kind regards,

Sathish Muthu

Academic Editor

PLOS ONE

Journal Requirements:

Additional Editor Comments:

Kindly address the concerns and comments of the reviewers as noted below

The manuscript demonstrates a high level of quality and possesses considerable research significance; however, there remain several issues that require resolution, specifically as outlined below:

1. Abbreviations in the tables should be completed.

2. Since the screening literature does not limit the study content, the included trials will vary greatly in terms of study purpose, clinical stage, disease type, etc., and there may be large heterogeneity in the study results. How to solve this problem? Also, please add limitations of this study to the discussion.

3. Please supplement the flow diagram for the selection process of articles.

Reviewers' comments:

Reviewer's Responses to Questions

Comments to the Author

1. Does the manuscript provide a valid rationale for the proposed study, with clearly identified and justified research questions?

Reviewer #1: Yes

Reviewer #2: Yes

Reviewer #3: Partly

2. Is the protocol technically sound and planned in a manner that will lead to a meaningful outcome and allow testing the stated hypotheses?

Reviewer #1: Yes

Reviewer #2: Yes

Reviewer #3: Partly

3. Is the methodology feasible and described in sufficient detail to allow the work to be replicable?

Reviewer #1: Yes

Reviewer #2: Yes

Reviewer #3: Yes

4. Have the authors described where all data underlying the findings will be made available when the study is complete?

Reviewer #1: No

Reviewer #2: Yes

Reviewer #3: Yes

5. Is the manuscript presented in an intelligible fashion and written in standard English?

Reviewer #1: Yes

Reviewer #2: Yes

Reviewer #3: Yes

6. Review Comments to the Author

You may also provide optional suggestions and comments to authors that they might find helpful in planning their study.

Reviewer #1: The authors have thoroughly addressed my feedback in this version, and I appreciate the effort put into the revisions. It looks well-polished and complete. Thank you for your work on this.

Reviewer #2: I have been able to read the revised version of the article. The authors have made a significant effort to improve the manuscript. I congratulate the authors for this novel proposal.

Reviewer #3: The manuscript demonstrates a high level of quality and possesses considerable research significance; however, there remain several issues that require resolution, specifically as outlined below:

1. Abbreviations in the tables should be completed.

Response: We have now defined these abbreviations. 

2. Since the screening literature does not limit the study content, the included trials will vary greatly in terms of study purpose, clinical stage, disease type, etc., and there may be large heterogeneity in the study results. How to solve this problem? Also, please add limitations of this study to the discussion.

Response: This review is focused on methodological studies of ACTs to identify methodological bias items specific to ACTs. We are not including studies based on disease populations and we are not searching for published interventional ACTs. We have further clarified in the methods. We added potential limitations in the discussion. 

3. Please supplement the flow diagram for the selection process of articles.

Response: We have added the PRISMA flow chart as a model for how we will select articles and search peer-reviewed databases and grey literature databases. 

7. PLOS authors have the option to publish the peer review history of their article (what does this mean?). If published, this will include your full peer review and any attached files.

Do you want your identity to be public for this peer review? For information about this choice, including consent withdrawal, please see our Privacy Policy.

Reviewer #1: No

Reviewer #2: Yes: M. Pardo Rios

Reviewer #3: No

---

## [Editor Report · Decision Letter 2]

12 Sep 2024

Methodological review to develop a list of bias items for adaptive clinical trials: Protocol and rationale

PONE-D-24-13528R2

Dear Dr. Staibano,

We’re pleased to inform you that your manuscript has been judged scientifically suitable for publication and will be formally accepted for publication once it meets all outstanding technical requirements.

Kind regards,

Sathish Muthu

Academic Editor

PLOS ONE

Additional Editor Comments (optional):

Congratulations to the authors for addressing the comments sufficiently to be considered for publication.
---

## [Editor Report · Acceptance letter]

24 Sep 2024

PONE-D-24-13528R2 

PLOS ONE

Dear Dr. Staibano, 

I'm pleased to inform you that your manuscript has been deemed suitable for publication in PLOS ONE. Congratulations! Your manuscript is now being handed over to our production team.

Kind regards, 

on behalf of

Dr. Sathish Muthu 

Academic Editor

PLOS ONE